# Compositional Features of Distinct Microbiota Base on Serum Extracellular Vesicle Metagenomics Analysis in Moderate to Severe Psoriasis Patients

**DOI:** 10.3390/cells10092349

**Published:** 2021-09-08

**Authors:** Chih-Jung Chang, Jing Zhang, Yu-Ling Tsai, Chun-Bing Chen, Chun-Wei Lu, Yu-Ping Huo, Huey-Ming Liou, Chao Ji, Wen-Hung Chung

**Affiliations:** 1Medical Research Center and Xiamen Chang Gung Allergology Consortium, Xiamen Chang Gung Hospital, Xiamen 361028, China; chan.chih.jung@gmail.com; 2Drug Hypersensitivity Clinical and Research Center, Department of Dermatology, Chang Gung Memorial Hospital, Linkou, Taoyuan 333423, Taiwan; chunbing.chen@gmail.com (C.-B.C.); c.Wei.lu@gmail.com (C.-W.L.); 3Department of Dermatology, The First Affiliated Hospital of Fujian Medical University, Fuzhou 350005, China; haozhangjinga@126.com; 4Department of Pathology, Tri-Service General Hospital, Taipei 114202, Taiwan; c909228@gmail.com; 5Cancer Vaccine and Immune Cell Therapy Core Laboratory, Department of Medical Research, Chang Gung Memorial Hospital, Linkou, Taoyuan 333423, Taiwan; 6College of Medicine, Chang Gung University, Taoyuan 333323, Taiwan; 7Whole-Genome Research Core Laboratory of Human Diseases, Chang Gung Memorial Hospital, Keelung 20445, Taiwan; 8Graduate Institute of Clinical Medical Sciences, Chang Gung University, Taoyuan 333323, Taiwan; 9Department of Dermatology, Xiamen Chang Gung Hospital, Xiamen 361028, China; hhyp123@adm.cgmh.com.cn (Y.-P.H.); liou_0309@163.com (H.-M.L.); 10School of Medicine, Shanghai Jiao Tong University, Shanghai 200240, China

**Keywords:** psoriasis, extracellular vesicle, microbiota, full-length 16S rRNA

## Abstract

The bacterial microbiota in the skin and intestine of patients with psoriasis were different compared with that of healthy individuals. However, the presence of a distinct blood microbiome in patients with psoriasis is yet to be investigated. In this study, we investigated the differences in bacterial communities in plasma-derived extracellular vesicles (EVs) between patients with moderate to severe psoriasis (PSOs) and healthy controls (HCs). The plasma EVs from the PSO (PASI > 10) (*n* = 20) and HC (*n* = 8) groups were obtained via a series of centrifugations, and patterns were examined and confirmed using transmission electron microscopy (TEM) and EV-specific markers. The taxonomic composition of the microbiota was determined by using full-length 16S ribosomal RNA gene sequencing. The PSO group had lower bacterial diversity and richness compared with HC group. Principal coordinate analysis (PCoA)-based clustering was used to assess diversity and validated dysbiosis for both groups. Differences at the level of amplicon sequence variant (ASV) were observed, suggesting alterations in specific ASVs according to health conditions. The HC group had higher levels of the phylum Firmicutes and Fusobacteria than in the PSO group. The order Lactobacillales, family Brucellaceae, genera *Streptococcus,* and species *Kingella oralis* and *Aquabacterium parvum* were highly abundant in the HC group compared with the PSO group. Conversely, the order Bacillales and the genera *Staphylococcus* and *Sphihgomonas*, as well as *Ralstonia insidiosa*, were more abundant in the PSO group. We further predicted the microbiota functional capacities, which revealed significant differences between the PSO and HC groups. In addition to previous studies on microbiome changes in the skin and gut, we demonstrated compositional differences in the microbe-derived EVs in the plasma of PSO patients. Plasma EVs could be an indicator for assessing the composition of the microbiome of PSO patients.

## 1. Introduction

Chronic plaque psoriasis or psoriasis vulgaris is the most common form of psoriasis and has clinical features characterized by recurrent episodes of red and scaly skin plaques located on the scalp, elbows, knees, umbilicus, and lumbar region [1]. Patients with psoriasis not only suffer from inflammatory skin diseases, but are also often burdened with comorbidities, such as hyperlipidemia, type 2 diabetes, obesity, inflammatory bowel disease (IBD), and cardiovascular disease [2]. Evidently, psoriasis is a systemic inflammatory disease, and it is crucial to develop new potential therapeutic strategies.

Extracellular vesicles (EVs) secreted by prokaryotes and eukaryotes are responsible for cell–cell communication [3]. The diverse cargos, size, and concentration of EVs reflect the state of their cells of origin and serve as biomarkers of various pathological conditions [4]. The lipid bilayer membrane of EVs from donor cells carries original contents, such as DNA, RNA (microRNA, mRNA, long noncoding RNAs), proteins, and lipids, which are subsequently deposited into the cytosol of recipient cells to trigger modulation of immunity [5]. Recently, EVs from immune cells have been shown to be involved in the pathogenesis of psoriasis. Many studies have shown that Th17 cells and the Th17 pathway play a more and more important role in the occurrence and development of psoriasis. As a downstream effector molecule of Th17 cells, interleukin-17A (IL-17A) can induce fibroblasts and epithelial cells to secrete a variety of cytokines and promote local inflammation [6]. Lin et al. demonstrated that the increased number of mast cells and neutrophils in psoriatic lesions contributes to the release of the pathogenic cytokine IL-17A through EV formation [7]. Claire et al. detected an increase in IL-17A-producing exosomes in psoriasis patients [8]. Drugs targeting free IL-17 and its receptor are nowadays the most effective treatment in managing psoriasis [9,10].The release of EVs is a bidirectional communication between keratinocytes and neutrophils in the inflammatory state of psoriasis [11,12]. Psoriatic keratinocytes induce the polarization of Th1/Th17 cells via specific microRNAs in secreted EVs and further promote psoriasis development [13]. In addition to immune-cell secreted EVs as pathogenic factors for psoriasis, encapsuled microRNA is considered a biomarker for psoriasis [14,15]. EVs from bacteria, called membrane vesicles (MVs) or outer membrane vesicles (OMVs) that originated from Gram-negative and Gram-positive bacteria, respectively, also contain microorganism-associated molecular patterns (MAMPs), such as nucleic acid, lipopolysaccharides (LPS), peptidoglycan, or toxin to modulate immunity [16]. Those MAMPs are recognized by different families of pattern recognition receptors (PRRs) expressed by both immune and nonimmune cells in the host and further trigger immunomodulatory signaling [17]. The EVs are characterized as transportation in circulation systems regardless of the EVs secreted from eukaryotes and prokaryotes. Dysbiosis of the skin or gut microbiota is a risk factor that contributes to inadequate innate and adaptive immunity [18,19]. The gut–skin axis is a critical factor in the pathogenesis of psoriasis. Microbe-derived EVs are detectable in the body fluids and play an important role in microbe–host interaction [20]. Pathogen-derived EVs have been demonstrated to regulate disease development, such as in neutrophilic pulmonary inflammation and atopic dermatitis [21]. Beneficial microbe-derived EVs have been found to ameliorate inflammation and affect distal host cells [22,23]. An in-depth analysis of EV metagenome could be an excellent means to systemically detect the bacteria–host relationship [24]. Studies have shown that the gut or skin microbiota play a critical role in psoriasis; however, no study has analyzed systemic microbiome factors. This study investigated the diversity and abundance of microbial EVs in sera collected from PSO patients and healthy controls (HCs). Full-length 16S metagenomic analysis identified biologically and statistically significant bacterial EV taxa as a diagnostic model.

## 2. Materials and Method

### 2.1. Sample Collection

Serum specimens were collected from PSO patients (PASI > 10) and HCs who had neither skin diseases nor any significant underlying diseases, such as metabolic syndrome, intestinal disease, and cancers. None of the participants used probiotics, systemic antibiotics, or steroids within 14 days prior to the study. All the participants provided written informed consent for the publication of their case details. The study was approved by the Ethics Committee of Xiamen Chang Gung Hospital (approval number XMCGIRB2020039, date of approval: 16 December 2020).

### 2.2. Isolation of EVs

Plasma EVs were isolated from blood samples via a series of ultracentrifugations. Blood samples at an amount of 4 mL were first diluted using fivefold PBS to decrease viscosity and centrifuged at 2000× *g* for 30 min at 4 °C to remove debris. The supernatant was transferred into new tubes and centrifuged at 10,000× *g* for 45 min at 4 °C. The supernatant was subsequently filtered using a 0.45 μm syringe filter (Millipore, Burlington, MA, USA) prior to ultracentrifugation at 100,000× *g* for 70 min at 4 °C (Optima L-100XP, Beckman Coulter, Brea, CA, USA). The exosome pellet was resuspended in 10 mL cold PBS, and the ultracentrifugation step was repeated. The final exosome pellet was resuspended in 100 μL of filtered PBS (0.22 μm) for subsequent analysis.

### 2.3. Transmission Electron Microscopy (TEM)

TEM with negative staining was used to examine the morphology of the isolated exosomes. Purified exosomes were diluted twofold to a volume of 10 μL and loaded onto a copper grid for 1 min, and the excess exosome solution was carefully removed using filter paper. The absorbed exosomes were stained with 10 μL 2% uranyl acetate for 1 min, and the excess fluid was removed using filter paper. The grids were dried for a few minutes at room temperature, and exosomal images were captured using TEM at 80 kV (HT7700, Hitachi High-Technologies Corporation, Minato, Tokyo, Japan).

### 2.4. Nano-Flow Cytometry (nFCM) Measurement 

The concentration and size of serum exosomes were analyzed using nFCM with a nanoanalyzer following the manufacturer’s instructions (N30E, nano-Analyzer, NanoFCM Inc., Xiamen, China). A 200 nm Silica nanosphere cocktail was used to calibrate the concentration and size of isolated serum exosomes. Twenty microliters of each sample was diluted using cold PBS (1:4 dilution ratio), and subsequently, 30 μL of each diluted sample was stained with 20 μL FITC mouse anti-human CD9 and FITC mouse anti-human CD81 antibodies (BD Biosciences, Franklin Lakes, NJ, USA) at 37 °C for 30 min. Isotype, IgG, is negative control for flow cytometry experiments (BD Biosciences, Franklin Lakes, NJ, USA). After incubation, the mixture was washed twice with PBS and centrifuged at 110,000× *g* for 70 min at 4 °C (CP100MX, Hitachi High-Technologies Corporation, Minato, Tokyo, Japan). The supernatant was discarded, and the pellet was resuspended in 50 μL cold PBS.

### 2.5. Western Blot Analysis

Protein concentrations of EVs were evaluated using a BCA Protein Assay Kit (Beyotime, Haimen, China) according to the manufacturer’s instructions. Bovine serum albumin (BSA) was used as a standard. Antibodies against EV markers of CD9 (ab92726, 1:1000, Abcam, Cambridge UK), CD63 (ab134045, 1:1000, Abcam, Cambridge UK), and calnexin (ab22595, 1:1000, Abcam, Cambridge UK) were tested against anti-mouse secondary antibody (400108, 1:5000, BioLegend, San Diego, CA, USA) using Western blotting. Protein lysate (10 μg per well) was loaded onto 10% or 15% SDS-PAGE gels depending on the molecular weight of the target proteins. The separated proteins were transferred onto a methanol-activated polyvinylidene fluoride (PVDF) membrane (Merck Millipore, Burlington, MA, USA). Membranes were blocked in a blocking buffer containing 5% skim milk powder in 1× TBST for 1 h at room temperature and then incubated with the primary antibodies for overnight at 4 °C. Excess primary antibodies were removed by washing with TBST three times. Next, the membranes were incubated with the secondary antibody for 1 h at room temperature, followed by incubation with an Immobilon™ Western Chemiluminescent HRP Substrate (ECL, Merck Millipore, Burlington, MA, USA) for 5 min at room temperature, for chemiluminescent signal detection using a ChemiScope 3300 mini (Clinx, Shanghai, China).

### 2.6. DNA Extraction from Plasma EVs

DNA derived from EVs was extracted following a method described previously [25] with slight modification. Bacteria and foreign particles in the EVs were eliminated using a 0.22 μm filter prior to DNA extraction. The EV samples were boiled at 100 °C for 30 min and centrifuged at 10,000× *g* for 30 min. The quality and quantity of DNA were measured using a NanoDrop assay. DNA concentration and purity were confirmed on a 1% agarose gel.

### 2.7. Library Preparation and Sequencing

TransStart^®^ FastPfu DNA Polymerase (TransGen Biotech, Beijing, China) was used for all PCR assays. For the full-length 16S rRNA gene sequencing, a specific primer set (27F: 5′-CCTACGGGNGGCWGCAG-3′, 1492R: 5′-GACTACHVGGGTAT CTAATCC-3′) was used according to the 16S metagenomic sequencing library preparation procedure (Pacific Biosciences, Menlo Park, CA, USA). The mixture of PCR products was separated using 2% agarose gel electrophoresis stained with SYBR green loading dye. PCR products were purified using a QIAquick Gel Extraction Kit (Qiagen, Hilden, Germany). Sequencing libraries were generated using a SMRTbell Template Prep Kit (Pacific Biosciences, Menlo Park, CA, USA). The quality of the library was assessed on a Qubit 4.0 fluorometer (Thermo Scientific, Waltham, MA, USA) and Femto Pulse system (Agilent, Santa Clara, CA, USA). Finally, the library was sequenced on a PacBio Sequel platform.

### 2.8. Processing and Analysis of Microbial Profile

The circular consensus sequence (CCS) reads were determined with a minimum predicted accuracy of 0.9, and the minimum number of passes was set to three in the official workflow of PacBio by using the SMRT Link software. After demultiplexing, the CCS reads were further processed with DADA2 (version 1.10.1) to obtain amplicons with single-nucleotide resolution. The DADA2 workflow includes quality filtering, dereplication, learning the dataset-specific error model, amplicon sequence variant (ASV) inference, and chimera removal. For each representative sequence, the feature-classifier and classify-consensus-blast algorithm in QIIME2 was employed to annotate the taxonomy classification based on the information retrieved from the NCBI database.

To analyze the sequence similarities among different ASVs, multiple sequence alignment was conducted by using the QIIME2 alignment MAFFT against the NCBI database [26]. Principal coordinate analysis (PCoA) was performed using the distance matrix to acquire principal coordinates for the visualization of sophisticated and multidimensional data. LEfSe applies LDA to bacterial taxa identified as significantly different and further assesses the effect size of each differentially abundant taxon. In this study, taxa with an LDA score (log 10) > 4 were considered significant. For functional analysis, functional abundances from 16S rRNA sequencing data were analyzed for the prediction of functional genes by using PICRUSt (v1.1.1) [27].

### 2.9. Statistical Analysis

Statistical analysis was performed using GraphPad Prism version 8.0 (GraphPad Software, San Diego, CA, USA). Data were shown as the mean ± SD. Differences between the two groups were assessed using an unpaired two-tailed Student’s *t*-test for EV size and alpha diversity. The Kruskal-–Wallis test was used to assess the significance of other differences in the distributions. Statistical significance was accepted at a two-sided *p* value of <0.05.

## 3. Results

### 3.1. Isolation and Characterization of EVs

Plasma EVs were isolated from the PSO and HC groups and identified and confirmed using TEM, nFCM, and Western blot analyses. Typical cup-shaped vesicles in EVs were detected using TEM (Figure 1A). The size profiles and concentration of nanoparticles were comparable between the HC and PSO groups (Figure 1B,C), average sizes of 82.79 ± 1.68 nm and 77.73 ± 3.40 nm in the HC and PSO groups, respectively (Figure 1B). After normalization, particle concentrations were determined to be (4.16 ± 1.38)10^9^ and (2.75 ± 1.29)10^9^ particle/mL for the HC and PSO groups, respectively. nFCM and Western blotting both were used to evaluate the purity of isolated EVs. EV protein markers (CD9 and CD81) were positive for isolated EVs compared with IgG as blank (Figure 1D). Western blot analysis confirmed the presence of exosomal markers, such as CD9 and CD63, whereas calnexin, as a non-EV marker, was not detected (Figure 1E). Taken together, the series of centrifugation processes was effective for the isolation of EVs.

### 3.2. Difference of Distinct Microbiota between Healthy Control and Psoriasis Patients

We characterized distinct microbiomes from the HC (*n* = 8) and PSO (*n* = 20) groups (Table 1). After removing unqualified sequences, a total of 666,677 raw reads and an average of 23,809 ± 8975 reads per sample were obtained. A total of 58,318 effective reads were generated, and each sample produced an average of 20,828 ± 7729 effective reads (range: 8844 to 36,078 effective reads). The rarefaction curve analysis suggested that the microbiome diversity was captured beyond 4000 reads across the whole sample as the number of observed species plateaued (Appendix A). The Venn diagram showed overlapping ASVs, indicating that healthy controls and patients had 448 and 829 ASVs, respectively, with 160 overlapping ASVs (Appendix A). Alpha diversity indices indicated significant differences in chao1, observed species, and PD whole tree between the two groups (*p* < 0.05, *p* < 0.05, and *p* < 0.0001, respectively), showing a higher abundance and diversity in the HC group (Figure 2A–C). The Shannon index was not significantly different between the two groups (Figure 2D). Beta diversity analysis was conducted to estimate the similarity of the microbiota community between the two groups. Unweighted UniFrac analysis and score plots of the partial least squares discriminant analysis (PLS-DA) were applied to compare the similarity between the two groups. The observed PCoA with two principal component scores accounted for 28.43% and 11.03% of the total variation, respectively (Figure 3A). The distances were clustered using UPGMA on unweighted UniFrac (Figure 3B). The other PLS-DA plots showed that there was a significant separation between the two groups, with 5.29% and 4.28% of total variation in PLS1 and PLS2, respectively (Figure 3C). A heat map of the top 35 species (Figure 3D) revealed significant differences between the microbial community profiles of the HC and PSO groups.

### 3.3. Distribution of the Predominant Bacteria at Different Taxonomic Levels

Taxonomically classified ASVs were associated with 13 phyla, 23 classes, 48 orders, 89 families, 180 genera, and 373 species. Predominant bacteria were identified by assessing the relative abundance of the top 10 bacteria at each taxon level (Figure 4). At the phylum level, Proteobacteria, Firmicutes, Bacteroidetes, and Actinobacteria accounted for the majority of the total sequences (Figure 4A and Appendix A). Firmicutes, Fusobacteria, and Cyanobacteria were significantly higher in the HC group (Appendix A, *p* < 0.01, Kruskal-Wallis test). At the class level, Betaproteobacteria, Gammaproteobacteria, Alphaproteobacteria, and Bacilli were predominant (Figure 4B, Appendix A), of which Bacilli and Epsilonproteobacteria were consistently more abundant in the HC group (Appendix A, *p* < 0.01, Kruskal-Wallis test). At the order level, Burkholderiales, Xanthomonadales, Neisseriales, Rhizobiales, Pseudomonadales, and Bacillales were predominant (Figure 4C and Appendix A). The relative abundances of Neisseriales and Bacillales were higher in the HC group (Appendix A, *p* < 0.05, Kruskal-Wallis test). At the family level, the predominant families were Burkholderiaceae, Xanthomonadaceae, Neisseriaceae, Alcaligenaceae, Moraxellaceae, and Staphylococcaceae (Figure 4D and Appendix A), of which Staphylococcaceae were more abundant in the PSO group, whereas Neisseriaceae were less abundant in the HC group (Appendix A, *p* < 0.05, Kruskal-Wallis test). At the genus level, *Luteimonas*, *Alcaligenes*, and *Kingella* were predominant (Figure 4E and Appendix A). *Kingella* was significantly higher in the HC group (Appendix A, *p* < 0.01, Kruskal-Wallis test). At the species level, the predominant species were *R. insidiosa*, *L. terricola*, and *K. oralis* (Figure 4F and Appendix A). Among them, the abundances of *R. insidiosa* and *K. oralis* were significantly different between the HC and PSO groups (Appendix A, *p* < 0.05, Kruskal-Wallis test). These results indicated distinctive differences in the bacterial communities of the HC and PSO groups.

### 3.4. Differential Analysis of Microbiota Composition for the PSO and HC Groups

LEfSe with a logarithmic LDA score cutoff >4.0 was used to compare the estimated phylotypes of microbiota from the PSO and HC groups. The microbial communities in the PSO group were significantly more diverse (Figure 5A). A histogram of the LDA scores indicated that the phylum Firmicutes, class Bacilli, order Lactobacillales, family Brucellaceae, genus *Streptococcus*, and species *K. oralis*, as well as *A. parvum*, were more abundant in the HC group, whereas in the PSO group, the order Bacillales, family Staphylococcaceae, genera *Staphylococcus* and *Sphihgomonas*, and species *R. insidiosa* were of greater abundance (Figure 5B).

### 3.5. Microbial Functional Properties Were Predicted Using PICRUSt

The prediction of genetic potential was analyzed by performing PICRUSt analysis based on 16S rRNA metagenome sequences. PICRUSt predicted metagenome content to Level 3 of the Kyoto Encyclopedia of Genes and Genomes database (KEGG) orthology and identified 268 functional pathways belonging to different Level 1 KOs. A total of 23 functional pathways were identified that belong to human diseases, metabolism, and organismal systems (Figure 6). Type 2 diabetes mellitus and pathways associated with cancer were enriched in the PSO group (Figure 6A). A variety of metabolic pathways, such as those of tryptophan metabolism, lipid biosynthesis proteins, and fatty acid metabolism, were significantly enhanced in the PSO group, while fructose and mannose metabolism and biosynthesis of ansamycins were increased in the HC group (Figure 6B). The PPAR, melanogenesis, and adipocytokine signaling pathways were significantly enhanced in the PSO group (Figure 6C). Collectively, our data suggest that a change in bacterial composition can lead to significant changes in gene functions, which correlates with inflammatory status in psoriasis.

### 3.6. Discussion

EVs are vesicle-like bodies that fall off the cell membrane or are secreted by the cell, with diameters ranging from 40 to 1000 nm. Claire et al. found that there was no significant difference in the mean size or number of serum EVs in patients with psoriasis compared with healthy controls, nor was there a significant difference in the mean size or number of serum EVs in patients with different severities of psoriasis [8]. This shows that all types of cells can release exosomes under physiological and pathological conditions. In this study, although we found that the size and concentration of SERUM EVs were statistically different between the HC group and the PSO group, this may be due to the small sample size affecting the statistical results.

It has been proved that EVs play a key role in the intercellular communication between host and symbiotic microorganisms [28]. In this study, we observed that the bacterial diversity and richness of the PSO group were lower compared with that of the HC group, which is similar to previous studies that reported lower α-diversity in the intestine and skin [29,30]. Our findings are consistent with the study by Huang et al. in that at the phylum level, intestinal samples from both the PSO and HC groups mainly comprised Proteobacteria, Actinobacteria, and Firmicutes [31]. Previous studies have shown that 7–11% of patients with inflammatory bowel disease (IBD) had psoriasis, and the changes in intestinal microorganisms were significantly related to the pathogenesis of IBD [32]. Proteobacteria, Actinobacteria, and Firmicutes have been shown to be the predominant microorganisms in the intestinal tract [33]. A previous study indicated that Proteobacteria and Actinobacteria were more abundant in the blood, whereas Firmicutes and Bacteroidetes in the gut [34]. According to the above results, we speculated that serum EVs could reflect some intestinal bacterial components.

*Bacillus* is a sporogenic Gram-positive anaerobic bacterium. *Clostridium difficile* and *B. cereus* are the main opportunistic pathogens in hospital infections. Lykke et al. reported that *Bacillus* contains functional homologs of CFR antibiotic-resistant genes, contributing to the widespread resistance associated with it [35]. Many studies have established the evidence that *Bacillus* can survive for decades under difficult environmental conditions by forming dormant spores, and can resist a variety of physical and chemical factors [36]. Recent studies have shown that *C. difficile* has a high infection rate in PSO patients with IBD [37]. The toxin secreted by these pathogens can bind to specific receptors on the intestinal epithelial cell membrane, causing mucosal damage and inflammatory reaction, leading to enteric diarrhea. Although there is no report indicating that Bacillus is associated with PSO, in the present study, we found that the abundance of Bacillus was significantly higher in the PSO group.

At present, studies have shown that psoriasis is related to the ecological microorganisms on the body surface. *S. aureus* is one of the most common Gram-positive pathogens causing skin and mucous membrane infections [38]. It can colonize the skin, triggering a variety of allergic inflammatory changes by secreting a variety of pathogenic factors and toxins, which affect the host’s immune system and homeostasis [39]. Evidence has shown that α-toxin secreted by *S. aureus* can act on keratinocytes, which is related to the severity of atopic dermatitis [40]. *S. aureus* can be detected in more than 50% of PSO patients, and its colonization is significantly related to the severity of psoriasis [41,42]. PSO patients are more likely to be infected with *S. aureus*, and there is a significant difference in the abundance of *Staphylococcus* between PSO patients and healthy individuals [43]. We found that the diversity and abundance of bacteria in the PSO group were lower compared with that in the HC group, while that of Staphylococcaceae was higher in the PSO group. Therefore, serum microbiome may have originated from the skin microecological system potentially.

Although a rare pathogenic bacterium, *Ralstonia* is usually detected in the intensive care unit and can be transmitted through polluted water [44,45]. Recent studies have demonstrated that the relative abundance of *Ralstonia* in the adipose tissue of obese patients is significantly higher than in healthy individuals [46]. Shanthadevi et al. found that the abundance of *Ralstonia* was also significantly increased in the intestines of obese patients with type 2 diabetes [47]. Metabolic diseases, such as diabetes and obesity, induced by a high-fat diet, are characterized by persistent low-grade inflammation [48]. Lipopolysaccharide (LPS) plays an important role in low-grade-inflammation-induced metabolic diseases [49]. *Ralstonia* can release LPS, which interacts with TLR4 to trigger inflammation. It has been confirmed that there is a common inflammatory pathway in the pathogenesis of type 2 diabetes and psoriasis [50]. In this study, the relative abundance of Ralstonia in the PSO group was significantly higher compared with that in the HC group, and enrichment of the type 2 diabetes pathway in PSO was confirmed using PICRUSt analysis.

PICRUSt analysis also confirmed that amino acid metabolism and other related functions were important functions of plasma EV microecology in PSO. Pathogenic microorganisms need to evolve through processes such as material and energy metabolism to adapt to changes in the body environment. Restricting fructose intake has been found to reduce the risks associated with tumor growth, progression, and outcomes of cancer treatment [51,52]. In this study, the fructose metabolic function of plasma EV microorganisms was enhanced in the HC group, while the cancer pathway was enhanced in the PSO group. The peroxisome proliferator-activated receptor (PPAR) is a member of the nuclear hormone receptor superfamily that plays an important role in the process of inflammation [53]. Since the pathogenesis of PSO involves epidermal keratinocyte hyperplasia and underdifferentiation as well as dermal vascular inflammation, the abnormal expression of PPAR may play a role. Westergaard et al. found that the expression of PPARβ was significantly higher in the lesions of PSO patients than those without lesions [54]. This study further demonstrated that the PPAR signaling pathway was more abundant in PSO patients. Adipose tissue can produce a variety of bioactive adipocytokines, such as leptin, adiponectin, resistin, and endolipids, and participate in the physiological and pathological processes, such as energy metabolism, insulin resistance, immunity, and inflammation. Adipocytokines may be an important link between psoriasis, obesity, and metabolic diseases, and are closely related to the occurrence and development of psoriasis [55,56], which was confirmed by the enhanced adipocytokine signaling pathway observed in the PSO group. Although our study shows that plasma EVs of microorganisms have great potential as diagnostic indicators for PSO, further clinical trials are needed to verify the suitability of the diagnostic model.

## 4. Conclusions

We analyzed microbe-derived EVs fromnormal individuals and patients with moderate to severe psoriasis in this study. In addition to the microbiome in the gut and skin, the alteration of bacterial diversity and community in microbe-derived EVs may reflect a specific pattern of the microbiome in psoriasis patients. The results could help clarify the role of the microbiome in psoriasis and could contribute to advances in diagnosis and treatment.

## Figures and Tables

**Figure 1 cells-10-02349-f001:**
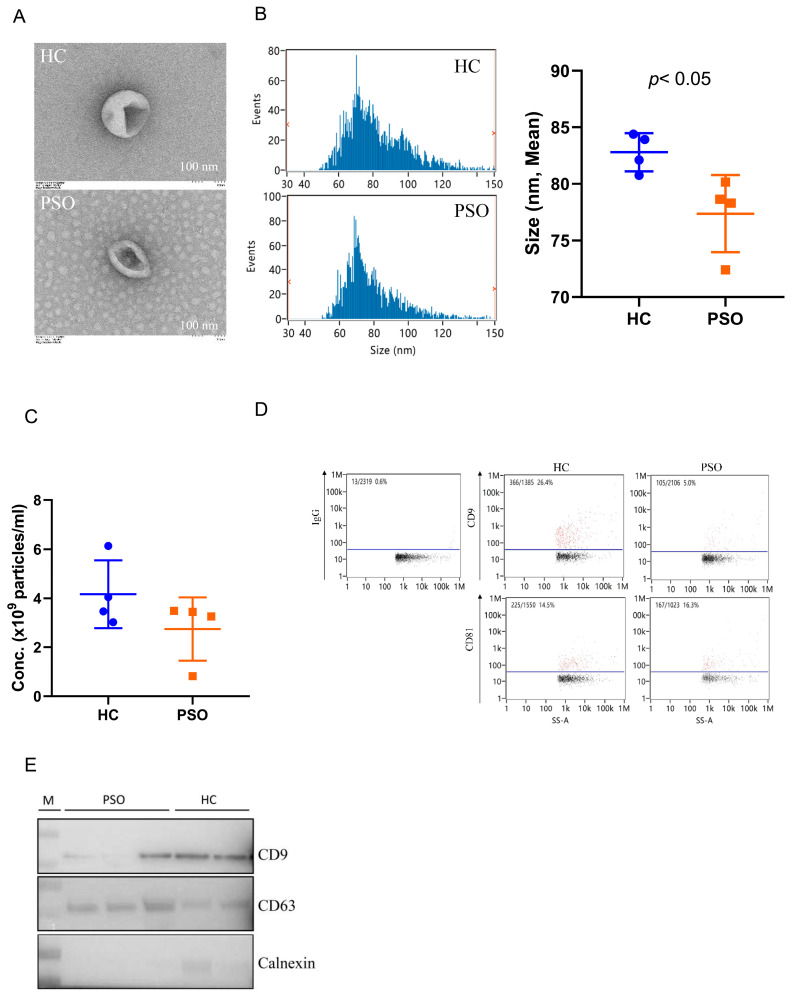
Isolation and identification of EVs from plasma. Electron micrograph of EVs was observed by transmission electron microscopy (bar = 100 nm) (**A**); nanoflow cytometry measurement (nFCM) was used to detect the diameter (**B**) and concentration (**C**) of EVs; nFCM was used to detect the expression of the surface markers CD9 and CD81 of EVS, and IgG was used as negative control (**D**); Western blot analysis to detect the expression of EV protein markers (CD9, CD63) and one non-EV marker (calnexin) (**E**) (HC: healthy control group; PSO: psoriasis group).

**Figure 2 cells-10-02349-f002:**
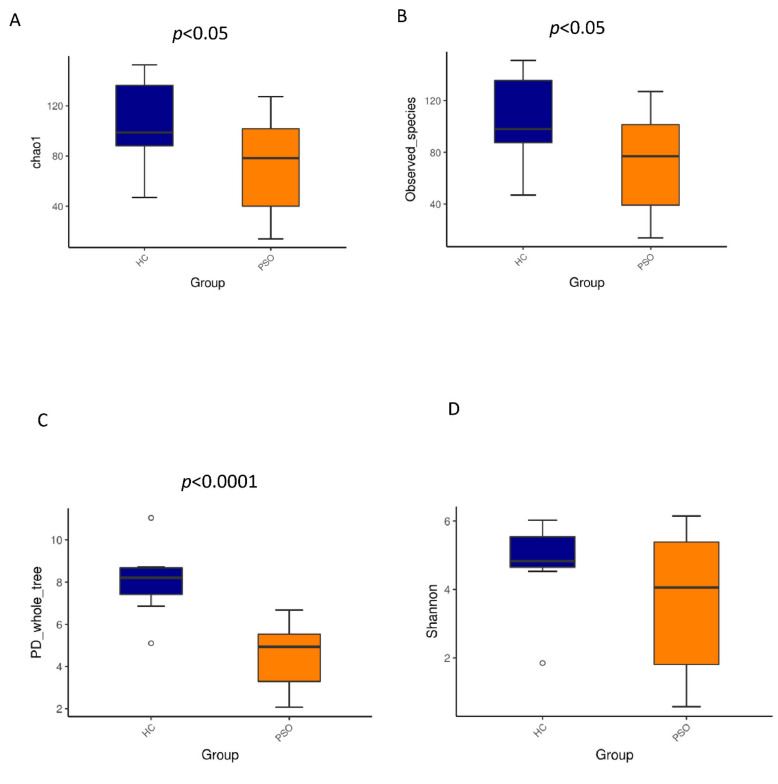
Alpha diversity metrics of plasma bacterial communities between healthy and psoriasis patients. Boxplots for comparison of species richness (chao1 index (**A**), observed species (**B**)) between the two study groups, (**C**) boxplots for comparison of phylogenetic diversity (PD whole tree), (**D**) boxplots for comparison of species diversity (Shannon index) (HC: healthy control group; PSO: psoriasis group).

**Figure 3 cells-10-02349-f003:**
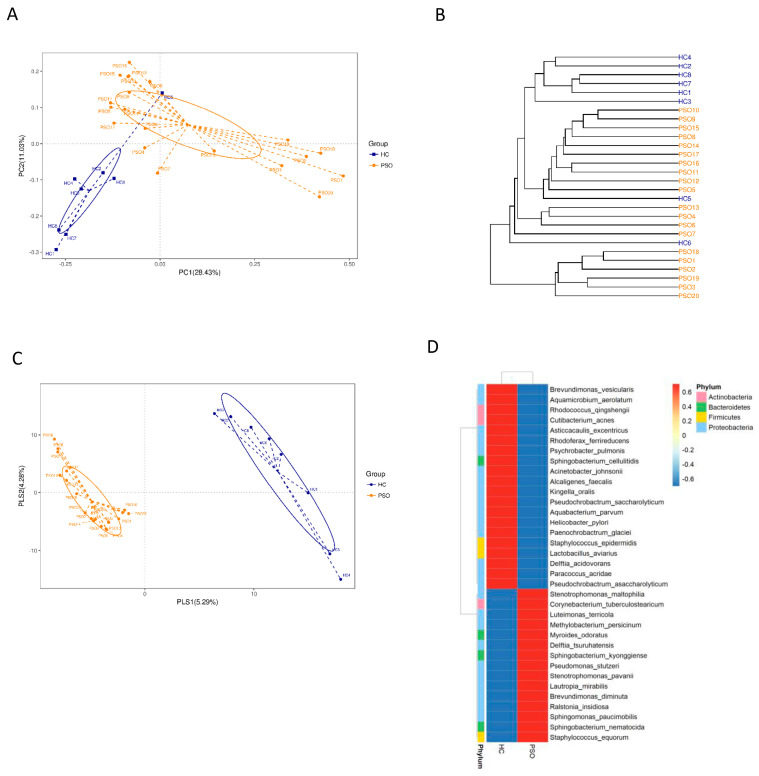
The β-diversity in distinct microbiota of healthy individuals and psoriasis patients. β-diversity changes in plasma microbiota across groups by the principal coordinate analysis (PCoA)(**A**), UPGMA cluster analysis (**B**), and partial least squares discriminant analysis (PLS–DA) (**C**). Each node represents each sample, and the control and PSO subjects are colored in blue and orange, respectively. Heat map of the top 35 genera among groups (**D**) (HC: healthy control group; PSO: psoriasis group).

**Figure 4 cells-10-02349-f004:**
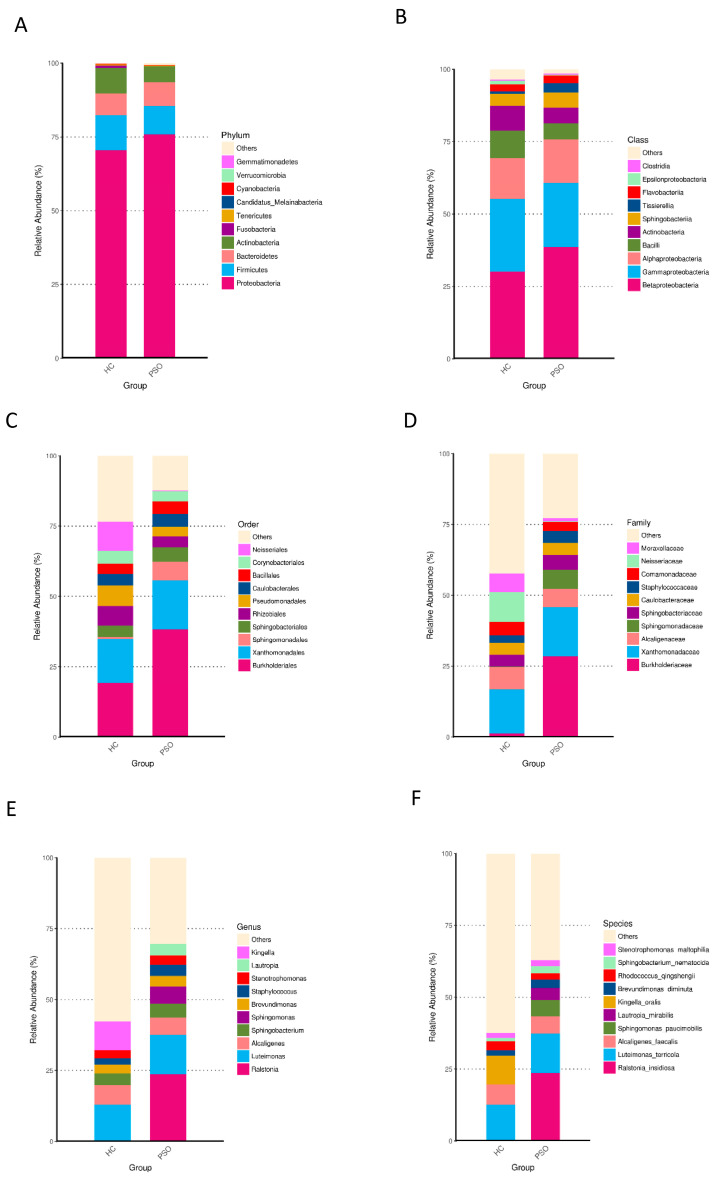
Relative abundance distribution of the top 10 species in phylum (**A**), class (**B**), order (**C**), family (**D**), genus (**E**), and species (**F**) in the two groups (HC: healthy control group; PSO: psoriasis group).

**Figure 5 cells-10-02349-f005:**
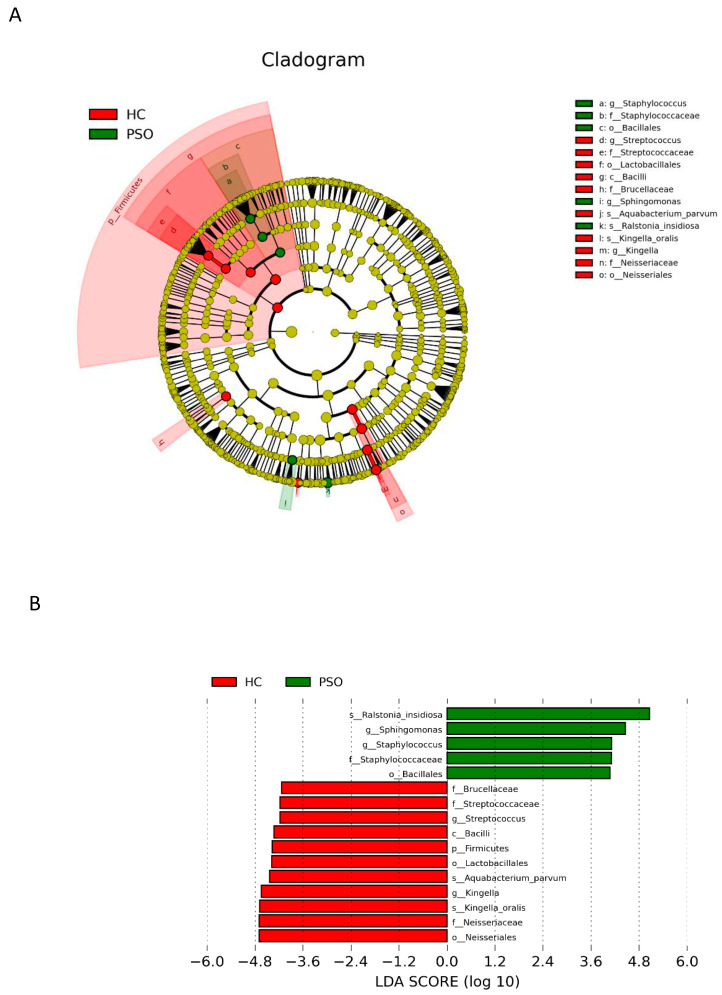
Analysis of species diversity between the two groups. Cladistic map of microbial evolution (**A**); histogram of LDA value distribution (**B**). In the evolutionary branching diagram, circles radiating from the inside out indicate taxonomic levels from phylum to genus (or species). Each small circle for a different taxon class represents the taxa of that class, and the diameter of the small circle is proportional to the relative abundance. The species with no significant differences are yellow, with red nodes representing the microbiota playing an important role in the control group and green nodes representing the microbiota playing an important role in the PSO group. The length of the histogram indicates the size of the impact of different species (LDA score) (HC: healthy control group; PSO: psoriasis group).

**Figure 6 cells-10-02349-f006:**
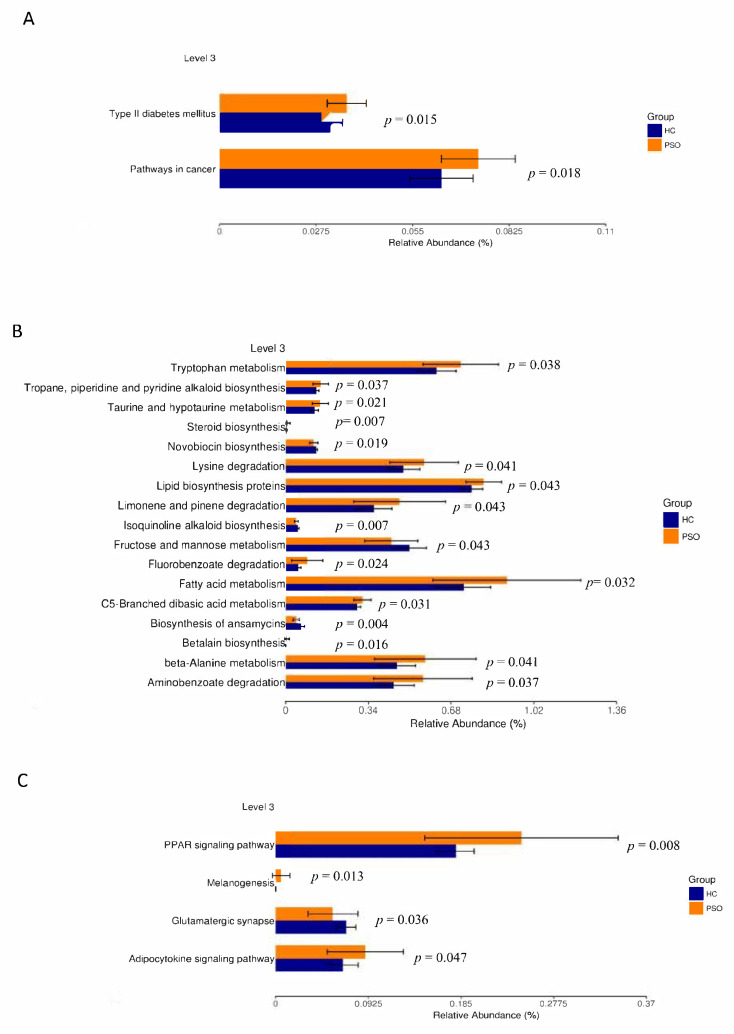
Mean relative abundance of predictive functions at KEGG Level 3 for the two groups. Pathways of human disease (**A**), pathways of metabolism(**B**), functional pathway of body system (**C**) (HC: healthy control group; PSO: psoriasis group).

**Table 1 cells-10-02349-t001:** Characteristics of psoriasis patients (PSO) and healthy controls (HC) studied in the discovery set.

Psoriasis Patients				Healthy Controls		
No.	Gender	Age	PASI	No.	Gender	Age
1	Female	47	29.4	1	Male	33
2	Male	35	15.2	2	Male	25
3	Male	67	11.3	3	Male	50
4	Male	37	15.7	4	Female	53
5	Male	33	15.6	5	Male	44
6	Female	26	10.2	6	Female	59
7	Male	34	11	7	Male	63
8	Female	50	10.6	8	Female	70
9	Male	63	13.2			
10	Male	31	38.4			
11	Female	30	14.5			
12	Male	66	12.6			
13	Male	88	10.8			
14	Female	43	13.5			
15	Male	50	12.2			
16	Female	28	11.8			
17	Male	44	12.6			
18	Female	24	11			
19	Male	40	19.2			
20	Male	53	11.5			

## Data Availability

The data presented in this study are available on reasonable request from the corresponding author.

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
