# Peer review of "Compositional Features of Distinct Microbiota Base on Serum Extracellular Vesicle Metagenomics Analysis in Moderate to Severe Psoriasis Patients"

_cells, 2021, doi:10.3390/cells10092349_

Round 1

Reviewer 1 Report

An interesting original study about the use of serum extracellular vesicle metagenomics analysis to differentiate patients affected by psoriasis and healthy controls; the results basically confirm the other studies already present in medical literature, showing a different population of microorganisms and lesser variability in psoriasis when compared to healthy controls;

I have some queries:

Authors should carefully check English, especially in the introduction paragraph; for example the first sentence: "Psoriasis is a chronic inflammatory dermatosis and approximately 2% people of worldwide " has no meaning.

The introduction should be expanded better describing psoriasis and making a small resume of treatment when you talked about anti-IL 17.

A big limitation, especially in the control groups, is the low number of patients in the study; a bigger sample would have allowed the results to be more significant..

page 2 line 60, you should add: "drugs targeting free IL17 and its receptor are nowadays the most effective treatment in managing psoriasis": and you should cite some articles such as: doi: 10.3390/healthcare9050543. and  doi: 10.1111/dth.13170.

In 2.9 statistical analysis please specify what type of t test you used (paired data? unpaired data?).

Also, a conclusion paragraph better highlighting the possible developments following this study would be a great addition to the study

Thank You

Author Response

1.Authors should carefully check English, especially in the introduction paragraph; for example, the first sentence: "Psoriasis is a chronic inflammatory dermatosis and approximately 2% people of worldwide " has no meaning.

Response: Many thanks for the reviewer's comment. We have checked and corrected the manuscript carefully and have proofread it with the assistance of a native speaker holding a professional background.

2. The introduction should be expanded better describing psoriasis and making a small resume of treatment when you talked about anti-IL 17.

Response: Many thanks for the reviewer's comment. We have added relevant content in the introduction according to your suggestions (Line 62-71).

3. A big limitation, especially in the control groups, is the low number of patients in the study; a bigger sample would have allowed the results to be more significant.

Response: Many thanks for the reviewer's comment. To collect important samples in this study, we recruited healthy serum samples without potential diseases (such as metabolic syndrome, intestinal diseases, and cancer) that may alter the production of extracellular vesicles and the microbial community. We added a description in line 101-102. Besides, in the 14 days before the study, none of the participants used probiotics, systemic antibiotics, or steroids. These criteria raised the difficulty for all sample recruitment. Accord to our results, the bacterial diversity, and community in extracellular vesicles were changed significantly between the two groups. Although the sample size is small, the results are worthy of reference.       

4. page 2 line 60, you should add: "drugs targeting free IL17 and its receptor are nowadays the most effective treatment in managing psoriasis": and you should cite some articles such as: doi: 10.3390/healthcare9050543. and  doi: 10.1111/dth.13170.

Response: Many thanks for the reviewer’s comment. We have added relevant content and cited relevant literature according to your suggestion (Line 70-71). 

5. In 2.9 statistical analysis please specify what type of t-test you used (paired data? unpaired data?).

Response: Many thanks for the reviewer’s comment. Differences were assessed using an unpaired two-tailed Student’s t-test that is described in line 192.

6. Also, a conclusion paragraph better highlighting the possible developments following this study would be a great addition to the study

Response: Many thanks for the reviewer’s suggestion. We’ve added descriptions ‘’We analyzed microbe-derived EVs in the paired normal and patients with moderate-to-severe psoriasis in this study. In addition to the microbiome in the gut and skin, the alteration of bacterial diversity and community in microbe-derived EVs may reflect the specific pattern of the microbiome in psoriasis patients. The results could help clarify the role of the microbiome in psoriasis and could contribute to advances in diagnosis and treatment.’’ in lines 425 to 429.

Reviewer 2 Report

Comments and Suggestions for Authors

Authors investigated the differences in bacterial communities in plasma-derived extracellular vesicles (EVs) between patients with moderate-to-severe psoriasis (PSO) and healthy controls (HC). The paper presented interesting data on microbacterial profile based on analysis of 16S rRNA gene sequence from plasma-derived EVs. For example, HC had higher levels of the species Kingella oralis and Aquabacterium parvum than in the PSO group. Conversely, the genera Staphylococcus and Sphigomonas, as well as Ralstonia insidiosa, were more abundant in the PSO group. The paper is good. However, the following points needs to be addressed.

  1. Introduction is quite enough information, but pleas clear more about the microbio-derived EVs.
  2. Table 1. (Pso) and PSO inconsistencies
  3. Can you explain why the exosome size is different between the HC and PSO groups in the discussion section?
    3. Author did not provide the volume of blood samples used to isolate EVs, how do you state the difference in EV number between different groups? Because the number of EVs is affected by the original volume of blood samples; therefore, it is an over-statement when authors said “the particle concentration of HC is slightly higher”
  4. It is an overstatement of “ Our study also confirmed that serum EVs could also be used to identify partially the bacterial composition in the intestinal tract.” when author just discus two articles identified Proteobacteria, Actinobacteria, Firmicutes, and Bacteroidetes in intestinal tract and blood that those of bacteria also detected in the study (line 328-333)

Author Response

  1. Introduction is quite enough information, but pleas clear more about the microbio-derived EVs.

Response: Many thanks for the reviewer’s comment. We’ve added descriptions ‘’EVs from bacteria, called membrane vesicles (MVs) or outer membrane vesicles (OMV) that originated from gram-negative and gram-positive bacteria respectively, also contain microorganism-associated molecular patterns (MAMPs) such as nucleic acid, lipopolysaccharides (LPS), peptidoglycan or toxin to modulate immunity[12]. Those MAMPs were recognized by different families of pattern recognition receptors (PRRs) expressed by both immune and non-immune cells in the host and further triggered immuno-modulatory signaling [13]. The EVs are characterized as transportation in circulation systems regardless of EVs secreted from eukaryotes and prokaryotes’’ according to your suggestion in line 76-84

  1. Table 1. (Pso) and PSO inconsistencies

Response: We’ve replace PSO with Pso as suggestion in table 1. (Line 239)

  1. Can you explain why the exosome size is different between the HC and PSO groups in the discussion section?

Response: Many thanks for the reviewer’s comment. We have supplemented the contents related to exosome size in the discussion section according to your suggestion (Line 340-348).

  1. Author did not provide the volume of blood samples used to isolate EVs, how do you state the difference in EV number between different groups? Because the number of EVs is affected by the original volume of blood samples; therefore, it is an over-statement when authors said “the particle concentration of HC is slightly higher”

Response: Many thanks for th reviewer’s comment. All plasma sample is 4 ml used for EVs isolation by a series of centrifugation (in line 107-108) and the concentration of EVs may be independent of plasma volume. Moreover, concentration of EVs could be reflected for status of pathological conditions. We’ve added the descriptions ‘’ The diverse cargos, size and concentration of EVs reflect the state of their cells-of-origin and serve as biomarkers of various pathological conditions’’ in lines 56-57.

  1. It is an overstatement of “ Our study also confirmed that serum EVs could also be used to identify partially the bacterial composition in the intestinal tract.” when author just discus two articles identified Proteobacteria, Actinobacteria, Firmicutes, and Bacteroidetes in intestinal tract and blood that those of bacteria also detected in the study (line 328-333)

Response: Many thanks for the reviewer’s comment. We have made changes to inappropriate descriptions in the discussion (Line 360-362).

Reviewer 3 Report

The paper is very innovative and interesting. Since many data and experiments have been done  the manuscript is not so easy to read and needs concentration and time. Anyway the data are useful for scientists in this area.

Author Response

The paper is very innovative and interesting. Since many data and experiments have been done the manuscript is not so easy to read and needs concentration and time. Anyway the data are useful for scientists in this area.

Response: Many thanks for the reviewer’s comments. To facilitate reading, we will draw a graphical summary and upload it.

Round 2

Reviewer 1 Report

The paper improved tremendously.

It is in my opinion publishable